# COVID-19 Underreporting in Brazil among Patients with Severe Acute Respiratory Syndrome during the Pandemic: An Ecological Study

**DOI:** 10.3390/diagnostics12061505

**Published:** 2022-06-20

**Authors:** Tainá Momesso Lima, Camila Vantini Capasso Palamim, Vitória Franchini Melani, Matheus Ferreira Mendes, Letícia Rojina Pereira, Fernando Augusto Lima Marson

**Affiliations:** 1Laboratory of Human and Medical Genetics, Postgraduate Program in Health Sciences, São Francisco University, Avenida São Francisco de Assis, 218. Jardim São José, Bragança Paulista, Sao Paulo 12916-900, Brazil; tainamomessolima65@gmail.com (T.M.L.); cvcpalamim@gmail.com (C.V.C.P.); vivismelani@gmail.com (V.F.M.); matheus.ferreira@mail.usf.edu.br (M.F.M.); leticiarogini@gmail.com (L.R.P.); 2Laboratory of Cellular and Molecular Biology and Bioactive Compounds, Postgraduate Program in Health Science, São Francisco University, Avenida São Francisco de Assis, 218. Jardim São José, Bragança Paulista, Sao Paulo 12916-900, Brazil

**Keywords:** COVID-19, SARS-CoV-2, severe acute respiratory syndrome, underreporting, undocumented, pandemic

## Abstract

Underreporting of Severe Acute Respiratory Syndrome Coronavirus 2 (SARS-CoV-2) infection is a global problem and might hamper Coronavirus Disease (COVID-19) epidemiological control. Taking this into consideration, we estimated possible SARS-CoV-2 infection underreporting in Brazil among patients with severe acute respiratory syndrome (SARS). An ecological study using a descriptive analysis of the SARS report was carried out based on data supplied by the Influenza Epidemiological Surveillance Information (SIVEP)-Flu (in Brazilian Portuguese, Sistema de Vigilância Epidemiológica da Gripe) in the period between January 2015 and March 2021. The number of SARS cases and related deaths after infection by SARS-CoV-2 or Influenzae was described. The estimation of underreporting was evaluated considering the relative increase in the number of cases with undefined etiological agent comparing 2020 to 2015–2019; and descriptive analysis was carried out including data from January–March/2021. In our data, SARS-CoV-2 infection and the presence of SARS with undefined etiological agent were associated with the higher number of cases and deaths from SARS in 2020/2021. SARS upsurge was six times over that expected in 2020, according to SARS seasonality in previous years (2015–2019). The lowest possible underdiagnosis rate was observed in the age group < 2 y.o. and individuals over 30 y.o., with ~50%; while in the age groups 10–19 and 20–29 y.o., the rates were 200–250% and 100%, respectively. For the remaining age groups (2–5 and 5–9 y.o.) underreporting was over 550%, except for female individuals in the age group 2–5 y.o., in which a ~500% rate was found. Our study described that the SARS-CoV-2 infection underreporting rate in Brazil in SARS patients is alarming and presents different indices, mainly associated with the patients’ age groups. Our results, mainly the underreporting index according to sex and age, should be evaluated with caution.

## 1. Introduction

Coronavirus Disease (COVID-2019) is caused by Severe Acute Respiratory Syndrome Coronavirus 2 (SARS-CoV-2) and presents a high contamination rate [1]. The COVID-19 pandemic has had great impact globally, since its development has generated a global health crisis, with consequent increase in unemployment rates, deaths that were directly or indirectly caused by the disease, health system overload, decreased gross domestic income, economic slowdown, and increased poverty, etc. [2,3,4,5].

The COVID-19 diagnosis gold standard is the SARS-CoV-2 Real Time Polymerase Chain Reaction (RT-PCR) [6,7], mainly for the early identification of cases (~7th day) of infection. Serology testing is also employed (immunochromatographic and enzyme-linked immunoassay-ELISA) aiming to detect antibodies, mainly after the 14th day of infection [6,7,8]. In Brazil, these tests were not made available for the whole population, and, in addition the tests show a normal error band that might result in false-negative or false-positive results. Both factors might contribute to disease underreporting and can cause increase in morbidity and mortality [6,9,10,11].

The number of cases and deaths from COVID-19 in Brazil is believed to be underreported mainly due to limitations to SARS-CoV-2 RT-PCR access [9,12]. These limitations in Brazil include: (i) difficulty in buying and high cost of inputs and equipment; (ii) low availability of equipment; (iii) number of professionals qualified to perform the RT-PCR; (iv) number of centers or laboratories qualified to perform the test; and (v) difficulty in transporting the material to centers or laboratories [9,10]. Moreover, the Brazilian regions with the highest percentage of unexpected deaths due to natural causes showed the least availability of RT-PCR and intensive care units (ICUs) during the pandemic, which might be associated with the unequal distribution of tests and ICU beds [12,13].

During the first weeks of the COVID-19 pandemic in Brazil, all suspected cases were tested with the SARS-CoV-2 RT-PCR. However, due to SARS-CoV-2 RT-PCR low availability, during the development of the pandemic only high-risk groups and more severe cases began to be tested. Consequently, the reporting of SARS-CoV-2 positive cases started to be possibly underestimated and undocumented [13,14,15]. Simultaneously, at the beginning of the pandemic, a higher number of admission of patients with severe acute respiratory syndrome (SARS) in hospitals was recorded. However, the number of COVID-19 positive cases described did not match the higher index of SARS cases. Therefore, the main problem might have been limited access to SARS-CoV-2 RT-PCR in Brazil, mainly in states in the North, Northeast and Midwest regions of the country [5,12,13,15].

The limited availability of tests to confirm COVID-19 diagnosis generates a scenery of uncertainty regarding the incidence and mortality rates owing to the disease in Brazil. Thus, relevant divergencies are created in the national epidemiological scenery, which hamper the implementation of successful public policies to control the disease. In such a context, it is necessary to become aware of the underreporting and how this can negatively impact the COVID-19 pandemic combat. Therefore, this study aimed to describe the epidemiologic profile of SARS patients in Brazil, according to etiological factor, age and sex, over a 5-year period, with the purpose of describing the possible underreporting of the SARS-CoV-2 infection.

## 2. Materials and Methods

This is an ecological study comparing the number of SARS patients with COVID-19, influenzae and undefined etiological agents in the 2020 period to those of the 2015–2019 period in order to arrive at an estimation of the underreporting rate of SARS-CoV-2 infection in Brazil. A descriptive analysis for SARS patients, also considering the data from January–March 2021, was also performed. This study was written following the STROBE recommendations.

The descriptive analysis of SARS (according to the World Health Organization, SARS is a syndromic diagnosis given to patients with the following criteria: fever (≥38 °C (100.4 °F)); one or more symptoms of lower respiratory tract infection: cough, difficulty breathing, dyspnea; radiological evidence of pulmonary infiltrates consistent with pneumonia or respiratory distress syndrome, or no alternative diagnosis that can fully explain the disease) patients’ data collection was carried out according to the markers found in the Oswaldo Cruz Foundation (FioCruz) electronic platform (info.gripe.fiocruz.br) for the last 5 years (2015–2020) [16], as well as for January–March 2021, whose data source is the Information System for Notifiable Diseases (SINAN; in Brazilian Portuguese: Sistema de Informação de Agravos de Notificação). The number of SARS cases and related deaths after infection caused by SARS-CoV-2 or Influenzae was described based on data supplied by the Influenza Epidemiological Surveillance Information (SIVEP)-Flu (in Brazilian Portuguese, Sistema de Vigilância Epidemiológica da Gripe). This platform reports cases of mandatory communication to the health authority, carried out by doctors, health professionals or those responsible for health establishments, public or private, regarding the occurrence of suspicion or confirmation of SARS [17].

Importantly, the SINAN was developed between 1990 and 1993, and regulated in 1998, providing for regular feeding into the national Brazilian database by municipalities, states and the Federal District mandatory. The SINAN system is mainly powered by the notification and investigation of cases of diseases and conditions that appear in the Brazilian national list of compulsory notification diseases. Therefore, SINAN covers all prevalent and relevant diseases such as SARS. However, as in any other system that restricts data input and therefore does not cover the total diversity of existing data, some bias regarding death information might occur. At the same time, false negative or positive results might also be reported mainly due to mistakes during data input.

The evaluation comprised total number of SARS cases according to the number of individuals affected and deaths from SARS, SARS caused by the SARS-CoV-2, SARS caused by Influenzae, and SARS with undefined etiological agent. The data were evaluated considering Brazilian geopolitical region (North, Northeast, Midwest, South, and Southeast), age group (years of age, y.o.: <2; 2–4; 5–9; 10–19; 20–29; 30–39; 40–49; 50–59; and +60), and sex (male or female). The data collection for descriptive analysis was extended up to one year after the announcement of the COVID-19 pandemic (March/2021).

The possible underreporting calculation was obtained through the difference between the total number of SARS cases and the sum of the SARS cases caused by Influenzae and by SARS-CoV-2, concomitantly to the sum of the average of cases with undefined etiological agent, weighted by the values obtained in the 5 years that preceded the pandemic: SARS–(SARS Influenzae + SARS COVID-19 + weighted average of SARS with undefined etiological agent in the period 2015–2019). The result of this operation possibly consists of the SARS cases due to SARS-CoV-2 infection that were not reported (estimation of undocumented cases of SARS COVID-19), either due to examination errors (false-negative, for example) or low availability of the SARS-CoV-2 RT-PCR test in certain places within the country. In addition, a calculation of the index of the relation between the estimation of the number of underreported SARS COVID-19 and the number of COVID-19 cases per sex and age group was carried out, aiming, describing the impact of these markers on the number of cases as, possibly, underestimated.

Statistical analysis was performed by the Statistical Package for the Social Sciences software (IBM SPSS Statistics for Macintosh, Version 27.0.) and OpenEpi software (OpenEpi: Open-Source Epidemiologic Statistics for Public Health, Version. Avaiable online: www.OpenEpi.com (accessed on 1 November 2021)) [18]. The chi-square statistical test was used to compare the proportion of the individuals with SARS due to SARS-CoV-2 or Influenzae infection or SARS due to undefined etiological agent, considering all patients’ features evaluated in this study (sex and age). The odds ratio (OR) with a 95% confidence interval (95% CI) was also calculated for each analysis carried out using the chi-square statistical test. OR was calculated using the OpenEpi software for 2 × 2 tables with the inclusion of the value for each patient characteristic. The study results were summarized in tables and figures. The figures were built up using the GraphPad Prism version 8.0.0 for Mac, GraphPad Software, San Diego, CA, USA, www.graphpad.com (accessed on 1 November 2021) [19].

## 3. Results

Figure 1A shows a SARS overview in Brazil from January 2015 to March 2021, considering SARS cases regardless of the etiological factor, and it was possible to identify five peaks of SARS cases preceding the COVID-19 pandemic. In 2020, there was a sharp increase in the number of SARS cases in all regions of the country due to the increase in COVID-19 cases. From 2020 onwards, the number of cases increased in all regions of the country, especially in the Southeast, the most populous region. In this region, cases soared in 2020, going from 5000 to over 11,000 notifications. Figure 1B demonstrates the annual seasonality of the Influenzae virus, which can be observed with the highest number of cases in 2016. In the second peak (approximately in May 2020), it is possible to visualize a sharp increase in Influenza cases, especially in the Southeast region. Curiously, in 2020 and early 2021, the lowest Influenzae rates were observed. Figure 1C presents the number of SARS cases with undefined etiological agent, with six peaks. Interestingly, it is possible to note that the figure is very similar to Figure 1A. Figure 1D shows the number of COVID-19 cases in 2020/2021, where two waves of the disease can be observed and the increased number of cases in all regions of the country from the beginning of the pandemic onwards. This figure also shows that the curves for each region have the same behavior pattern, i.e., except for the Southeast, the most populous region with greater global integration and therefore with greater variation, incidence in the different regions of the country follows the same trend.

Figure 2A presents the overview of number of deaths from SARS in Brazil during the period from January 2015 to March 2021, showing death cases regardless of etiological factor. In 2020, a sharp increase in the number of deaths from SARS was observed in all regions of the country due to the increased number of COVID-19 cases. The Southeast region presented the highest mortality rate in 2020, followed by the Northeast and North regions, respectively. Figure 2B demonstrates the annual seasonality of the Influenzae virus, with the highest number of deaths in 2016. Figure 2C shows the number of deaths from SARS with undefined etiological agent, with a major peak in 2016 and a sharp increase in 2020–2021. The Southeast region predominates in May 2016 and a trend towards low number of deaths is observed (still predominantly in the Southeast region) with three subsequent peaks. From May 2021 onwards, the figure shows a sharp rise in the number of deaths with the highest peak observed in the North region, which suffered from health service collapse (100% occupation), including lack of oxygen. Figure 2D presents the number of deaths from COVID-19 in 2020/2021, where two waves of the disease are observed, along with a high number of cases in all regions of the country from the beginning of the pandemic onwards. The data shown in Figure 1 and Figure 2 demonstrate the SARS evolution according to the Brazilian geopolitical regions.

Figure 3 demonstrates the increase in the number of cases (Figure 3A) and deaths (Figure 3B) caused by SARS in 2020/2021 due to the evolution of the COVID-19 pandemic, simultaneously with the increase in the number of SARS cases with undefined etiological agent. In the period 2020/2021, an increase of ~6 times in SARS cases in relation to the period 2015-2019 was described. These Figure 3A,B also reveal an association between the SARS curve and the SARS curve with undefined etiological agent—in 3B a peak is more clearly observed when notifications by COVID-19 agents and influenzae declined, but the number of deaths from SARS peaked at the expense of deaths from SARS with undefined etiological agent, thus translating into a scenario of possible underreporting.

Figure 4 presents the number of cases in absolute value with possible COVID-19 underreporting, according to sex and age (Figure 4A), and according to age group only (Figure 4B). Male individuals presented a higher absolute value of possible underreporting and both sexes showed an increase in possible COVID-19 underdiagnosis in older age groups. Therefore, cases involving male individuals that were over 60 y.o. represented higher possible underreporting in absolute value.

Figure 5 shows the relative number (percentage) between SARS cases with possible COVID-19 underdiagnosis, and COVID-19 confirmed cases according to sex and age groups. The lowest underdiagnosis rate occurred in the <2 years’ group and individuals over 30 y.o., with a ~50% rate. In the age groups 10–19 and 20–29 years, the possible underdiagnosis rates were 200–250% and ~100%, respectively. For the remaining age groups (2–5 and 5–9 years), the underdiagnosis rate was over 550%, except for female individuals in the 2–5 age group, whose observed value was ~500%. The data are presented per sex and age (Figure 5A) and per age only (Figure 5B).

Table 1 describes the profile of SARS patients in Brazil in 2020 according to their sex and age group. The total number of people affected by SARS was 460,107, out of which 194,974 were female and 265,133 were male. Out of that total 275,165 had COVID-19; 1762 had Influenzae virus infection; and 183,180 were in the undefined etiological factor group. The population affected by SARS increased according to the increase in age and the same occurred with COVID-19 cases. The group of patients that presented the highest number of individuals affected by the SARS-CoV-2 was that of male individuals that were over 60 y.o. (76,823 cases), while the least affected groups were those <2 y.o. and female (657 cases). The profile described for the most affected group was the same as that for the undefined etiological agent group (42,816 individuals).

Figure 6 also presents the OR and its 95% CI for the association between the SARS caused by undefined etiological agent compared with the presence of SARS due to COVID-19 or Influenzae virus infection. In this case, we could observe that female individuals are more prone to be classified as SARS due to undefined etiological agent when compared with male patients. We could also see that all patients’ age groups, except for patients in the 50–59 y.o. group, were more prone to be classified as SARS due to undefined etiological agent when compared with patients aged +60 y.o.

## 4. Discussion

SARS is a zoonotic respiratory infectious disease that affects humans and presents high morbidity and mortality rates [1], and was identified in Brazil many years ago. However, from 2020 until now, (a year after the start of the COVID-19 pandemic), an increasing trend has been observed in the number of cases of this disease. This sharp peak supports the hypothesis that the increased number of SARS cases is due to COVID-19; noteworthy, the Influenzae cases were almost zero during the pandemic, since social isolation was sufficient to reduce its spread [20]. However, the COVID-19 cases reported are not enough to justify the SARS increased number. Hence, this study analyzed and described the profile of SARS patients in Brazil with the purpose of describing the possible underreporting of the SARS-CoV-2 infection.

In this study, underreporting was calculated through the following operation, based on data collected on the database SIVEP-Flu: SARS–(SARS Influenzae + SARS COVID-19 + weighted average of SARS with undefined etiological agent in the period 2015–2019). Nevertheless, some studies found in the literature calculated the underreporting through the ratio of the case fatality rate (case-fatality ratio (CFR)-basal CFR) using the lethality rate of observed COVID-19 cases. Basal CFR was defined as the number of deaths divided by the number of cases of the disease, and estimates of CFR were used through the confirmed cases and the lethality observed of COVID-19 [21]. Another study in the literature calculated the underreporting only with the basal CFR, using published estimates of the baseline CFR, adjusted for delays, then calculating the ratio of this baseline CFR to an estimated local delay. In this study, a Bayesian Gaussian process model was used to estimate the disease underreporting temporal pattern [22].

It seems relevant to emphasize that underreporting might present a differentiated impact according to the individuals’ age group and sex, as well as a result from false negatives, as mentioned before, and the low availability of diagnostic tests, mainly RT-PCR [12]. In Brazil, the increase in the number of COVID-19 cases and the consequent admission to hospitals were inversely proportional to the availability of diagnostic tests. Laboratory tests demand time for analysis and test kits were scarce in the period, which hampered the supply according to the affected population’s demand.

Therefore, at the beginning of the COVID-19 pandemic, the majority of cases were tested, but as the pandemic evolved there were not enough tests available for all the affected population. This fact also affects the results of this study. For example, our Section 3 reports that SARS cases regardless of the etiological factor demonstrated similarities with the number of SARS cases with undefined etiological agent. In this context, SARS notifications seem to follow the trend of being notifications with unspecified etiological agent, possibly resulting in COVID-19 underreporting, and this similarity follows the same pattern for each Brazilian region. Hence, the most commonly affected groups were prioritized for testing: male individuals over 60 y.o., as shown in the Results Section 3. Maybe, the least affected groups (<2 y.o. and female) were less commonly tested and therefore more underreported, mainly the age group 2–9 y.o. (>500% underestimated cases). This finding was observed in all Brazilian geopolitical regions, but more significantly in the Midwest and North regions, i.e., the regions with fewer available SARS-CoV-2 RT-PCR tests.

Curiously, other studies developed in Brazil reported similar results, with high levels of underreporting [20,21,23,24,25]. Nevertheless, that is a problem present in several other countries, such as Peru, Iran, United Kingdom, and India [22,24,26]. According to one study which used cases and fatalities of COVID-19 all over the world, in Peru, for instance, in a period of three months in 2020 (between 1 April and 1 July 2020), there were 690% excess deaths when compared to confirmed COVID-19 deaths and 3396 reported COVID-19 deaths per 100,000 cases, whilst in the United Kingdom there were 199% excess deaths, and 23,642 reported COVID-19 deaths per 100,000 [22].

Moreover, the SARS-CoV-2 RT-PCR might present false-negative results due to several factors such as laboratory mistakes and low genetic material availability [11,12,21]. False negative results present serious implications for isolation and the risk of transmission by affected individuals, thus, being associated with the possibility of underreporting [27]. In addition, different report rates are observed when Brazilian states are compared [21].

In this context, underreporting becomes a problem as it goes against pandemic coping strategies. The absence of figures showing the real dimension of the health crisis makes it difficult to approach the government and the private sector to address this health issues, and a lack of theoretical basis for planning is observed from the municipal to the federal levels. The allocation of resources is impaired both qualitatively and quantitatively when those in charge do not know where and when to invest. Thus, ICU bed reservations, mass tests, restriction or not of the circulation of people and commerce are some of the main approaches that are harmed by COVID-19 underreporting [5,9,12,13]. Noteworthy, this study focuses on underreporting among those with SARS, but the underreporting rate is probably much higher when considering all the Brazilian population, in which the diagnostic tests provided by the Brazilian Unified Health System (SUS, in Brazilian Portuguese–Sistema Único de Saúde) are extremely limited, and the tests provided by the private sector are expensive and unaffordable for most of the Brazilian population. Another issue in Brazil is that ICU beds are divided unequally between the private and public health systems, as well as among states and the Federal District. Many patients who depend on the SUS could not be properly assisted, due to the lower availability of ICU beds in the public system when compared to private institutions [13]. With the development of the COVID-19 pandemic and the late involvement of the government and health authorities in dealing with the situation, the problem of lack of beds worsened [13]. After a year, the economy and the health system of the country collapsed: patients from both the private and public health systems could not be properly assisted due to lack of oxygen cylinders, for example, which occurred in Manaus and that possibly resulted in a high number of deaths in the northern region [28]. In fact, in Brazil, ICU beds are limited and do not support the whole population that require this service currently; consequently, these individuals are probably underreported.

The Brazilian federal government took too long to recognize the severity of the disease, presenting official sources with contradictory data referring its impact and encouraging the end of the quarantine/isolation period [2,9,14,29]. This fact supports the hypothesis that underreporting has occurred since the beginning of the pandemic, pointing out that the real number of affected individuals and deaths in Brazil might have been much higher than that officially reported [2,9,14] and, therefore, the disease dissemination might have been underestimated. Moreover, some fake news and the denial of part of the Brazilian population in relation to the pandemic were problems that worsened the number of cases and deaths [3,9].

After a year as of the start of the pandemic, Brazil experienced a peak and the worst moment regarding the number of cases and deaths, beyond the mark of 4000 deaths per day. The absence of an effective control system during and after the lockdown period, only resulted in losses for the economy and damage to the population’s physical and mental health, in addition to the collapse of the health system [14]. This scenery portrays lack of assistance to the population, lack of public policies to detain the disease, and the consequent increase in underreporting. Consequently, this created a generalized demoralization and decrease in the population’s trust in the government, mainly at the federal level [2,3,5,30].

The development of vaccines brought some hope, but there is still the challenge of responding to the constant evolution of the pathogen in order to reach immunity [31,32,33,34]. Moreover, until high levels of protection through vaccination are reached globally, protective and diagnostic measures must be kept.

Another challenge observed is the capability of this disease to provoke other conditions such as pneumonia and SARS, which can be characterized as the main causes of death. Therefore, a subjective bias is seen, since medical doctors cannot confirm or deny that the death was caused by COVID-19 based on clinical data only, without laboratory examinations. In addition, some deaths from COVID-19 complications might have been caused by assistance delay (death at home) and respiratory insufficiency in the hospital due to lack of oxygen. In such cases, a COVID-19 related death report is not generated, but other causes are reported, [21,35,36] for example, SARS resulting from undefined etiological agent.

Some limitations of this study regard the fact that the data was obtained from a system fed by the SIVEP-Flu, which restricts data input and diversity such as the inclusion only of Influenzae virus as etiological agent for SARS; a bias regarding reporting and data input might occur; loss of data due to false-negative as a function of the collection data and the possibility of admission in hospital due to COVID-19 not being entered into the system due to lack of characteristic respiratory symptoms, and thus being recorded as SARS; and a bias regarding the non-distinction of SUS and private health systems in the data obtained from SIVEP-Flu, which prevented a better characterization of the profile of COVID-19 patients.

Although the national surveillance guide has broadened the reporting criteria, including cases without fever, the most restrictive filter (presence of fever) was used to list our data, thus, we could maintain compatibility with SARS cases based on the international definition, without creating divergence within the study, since at the beginning of the pandemic cases without fever were not included.

## 5. Conclusions

The SARS-CoV-2 infection possible underreporting rate in Brazil among SARS patients is alarming and presents different indices, mainly associated with the patients’ sex and age group, which increased with the evolution of the COVID-19 pandemic due to the lack of tests. Therefore, this is the cause of the impressive underreporting in 2–9 y.o., and maybe the least affected group is the least tested, and so the most underreported. Thereby, a publicly accessible portal should be developed with more realistic and reliable data on the COVID-19 pandemic, in order to undo contradictions in official data, create public policies, and guide the population.

## Figures and Tables

**Figure 1 diagnostics-12-01505-f001:**
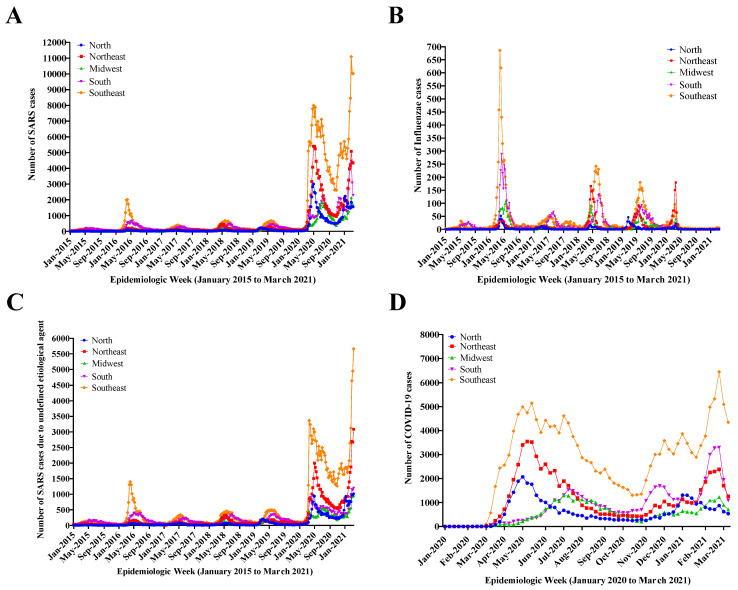
Severe Acute Respiratory Syndrome (SARS) overview in Brazil during the study period (January 2015 to March 2021). (**A**) SARS cases regardless of the etiological factor. (**B**) Evolution in the number of SARS cases resulting from Influenzae. (**C**) Evolution in the number of SARS cases with undefined etiological agent. (**D**) Evolution in the number of Coronavirus Disease (COVID-19) cases in 2020–2021.The data presented in this figure demonstrate SARS evolution according to Brazilian geopolitical regions. In Figure 1A–C, the description of the epidemiological weeks from Jan 2015 to Mar 2021 is observed. In Figure 1D, epidemiological weeks from Jan 2020 to Mar 2021 are described.

**Figure 2 diagnostics-12-01505-f002:**
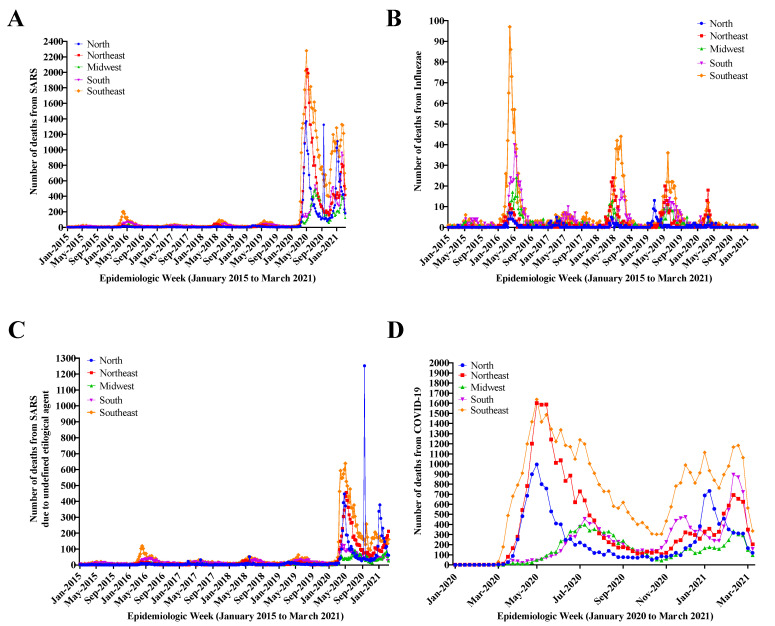
Death from severe acute respiratory syndrome (SARS) overview in Brazil during the study period (January 2015 to March 2021). (**A**) Deaths from SARS regardless of etiological factor. (**B**) Evolution in the number of deaths from SARS resulting from Influenzae. (**C**) Evolution in the number of deaths from SARS with undefined etiological agent. (**D**) Evolution in the number of deaths from Coronavirus Disease (COVID-19) in 2020–2021. The data presented demonstrate the SARS evolution according to Brazilian geopolitical regions. In Figure 1A–C, the description of the epidemiological weeks from Jan 2015 to Mar 2021 is presented. In Figure 1D, the epidemiological weeks from Jan 2020 to Mar h2021 are described.

**Figure 3 diagnostics-12-01505-f003:**
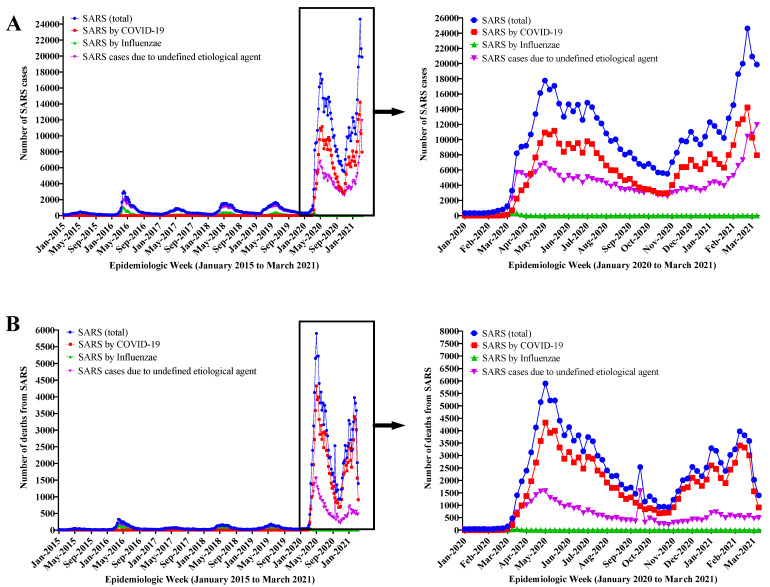
Number of cases (**A**) and deaths (**B**) from severe acute respiratory syndrome (SARS) due to the Coronavirus Disease (COVID-19) pandemic development, according to the etiological agent, in the period from January 2015 to March 2021.

**Figure 4 diagnostics-12-01505-f004:**
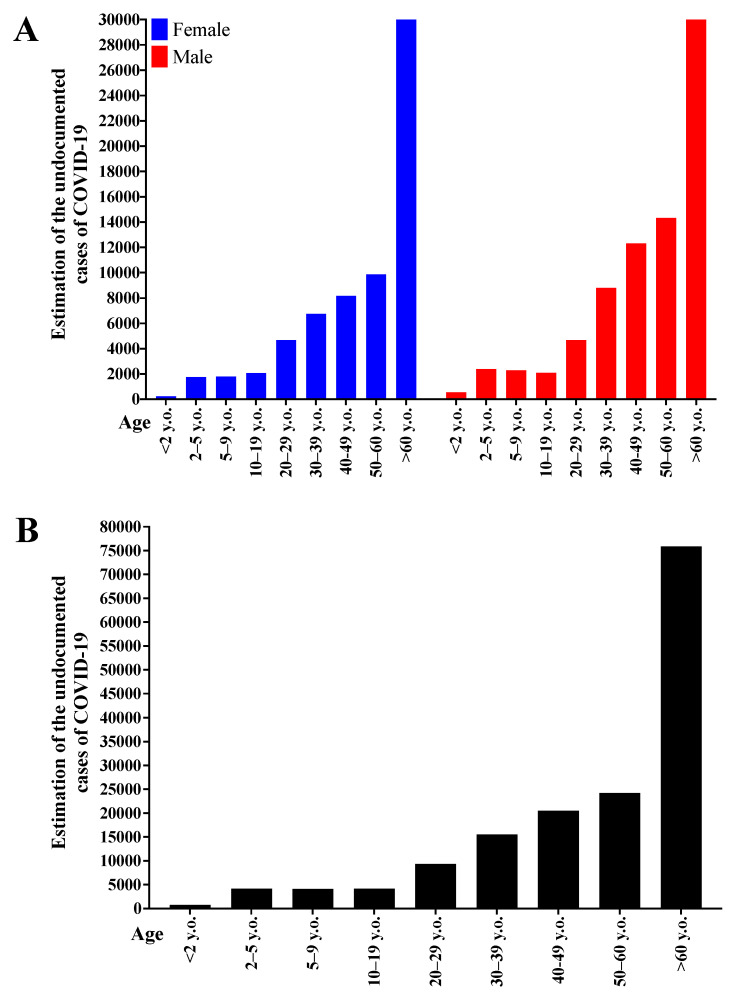
Number of cases in absolute value of possible Coronavirus Disease (COVID-19) underdiagnosis according to sex (female and male) and age groups (years old, y.o.: <2; 2–4; 5–9; 10–19; 20–29; 30–39; 40–49; 50–59; and +60) (**A**) and according to age only (**B**). The analysis included only data from 2020.

**Figure 5 diagnostics-12-01505-f005:**
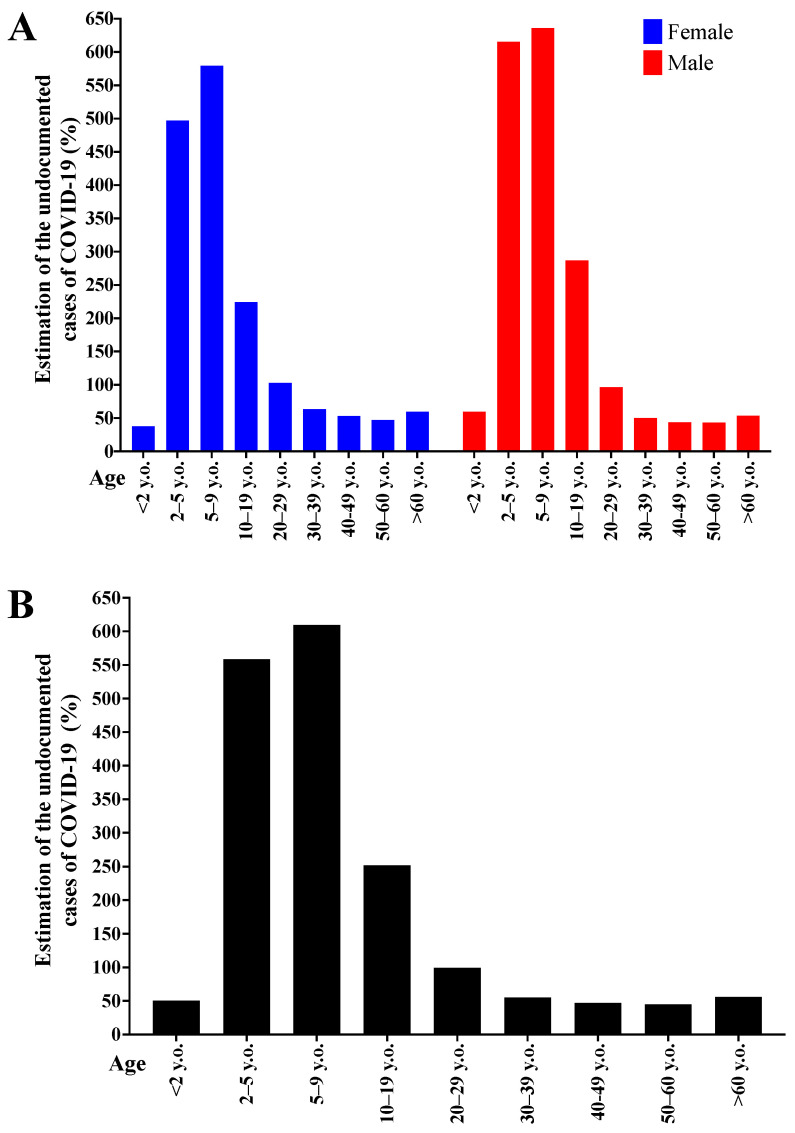
Relative number comparing the number of severe acute respiratory syndrome (SARS) cases with Coronavirus Disease (COVID-19) underdiagnosis and the number of COVID-19 cases per patients’ sex and age group. The data are presented according to sex and age (**A**) and according to age only (**B**). The analysis included only data from 2020.

**Figure 6 diagnostics-12-01505-f006:**
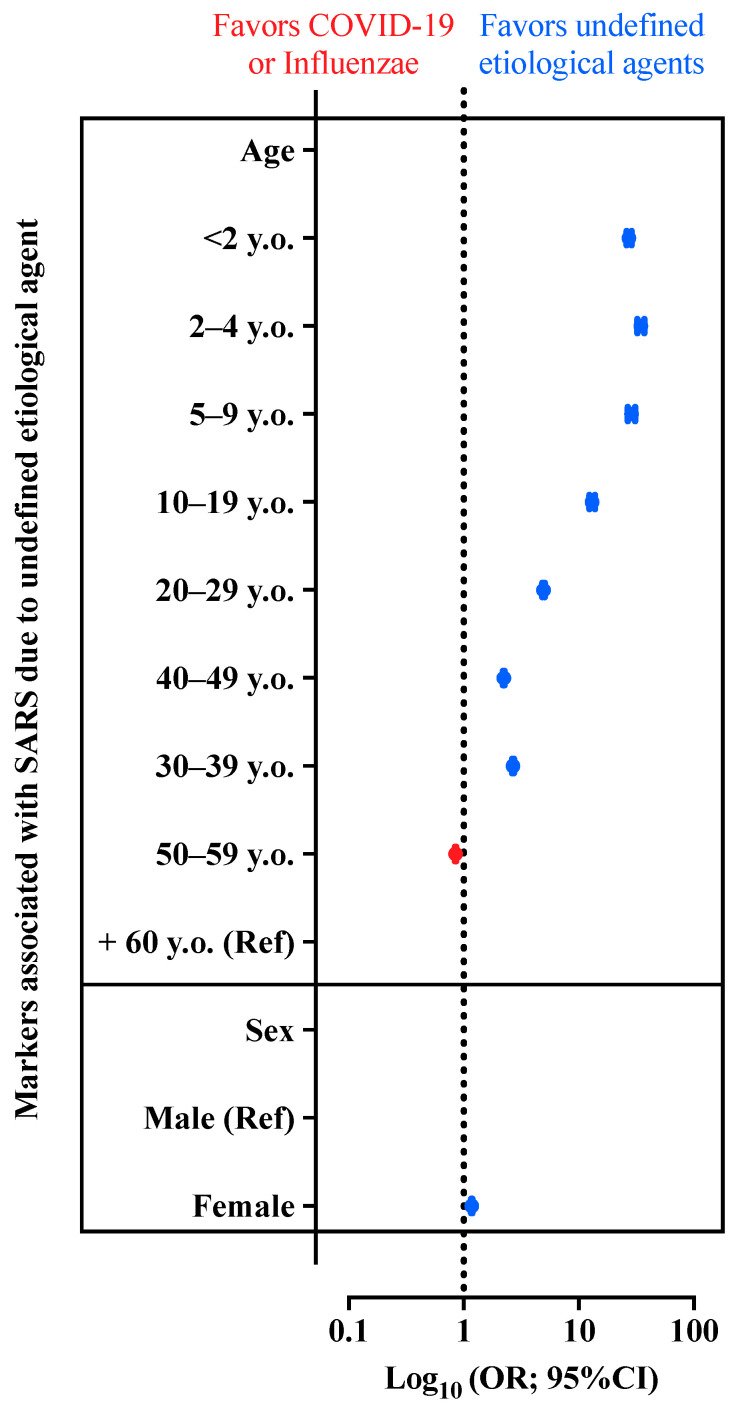
Odds ratio (OR) and 95% confidence interval (95% CI) for the association between patients with severe acute respiratory syndrome (SARS) due to undefined etiological agent compared with patients with SARS due to Coronavirus Disease (COVID-19) or Influenzae virus infection regarding sex and age. Ref, reference.

**Table 1 diagnostics-12-01505-t001:** Profile of severe acute respiratory syndrome (SARS) in 2020.

Age Group	Sex		Total
SARS	Female (%)	Male (%)	
<2 years old	5514 (42.13%)	7575 (57.87%)	13,089
2–4 years old	3500 (45.26%)	4233 (54.74%)	7733
5–9 years old	2782 (45.00%)	3395 (55.00%)	6177
10–19 years old	3601 (51.78%)	3353 (48.217%)	6954
20–29 years old	10,065 (49.95%)	10,086 (50.05%)	20,151
30–39 years old	18,301 (40.35%)	27,053 (59.65%)	45,354
40–49 years old	24,356 (37.14%)	41,220 (62.86%)	65,576
50–59 years old	31,621 (39.53%)	48,376 (60.47%)	79,997
+60 years old	95,234 (44.28%)	119,842 (55.72%)	215,076
General total	194,974 (42.38%)	265,133 (57.62%)	460,107
**Coronavirus Disease (COVID-19)**	**Female (%)**	**Male (%)**	**Total**
<2 years old	657 (40.90%)	949 (59.10%)	1606
2–4 years old	355 (47.65%)	390 (52.35%)	745
5–9 years old	310 (46.13%)	362 (53.87%)	672
10–19 years old	925 (55.89%)	730 (44.11%)	1655
20–29 years old	4555 (48.46%)	4845 (51.54%)	9400
30–39 years old	10,637 (37.78%)	17,521 (62.22%)	28,158
40–49 years old	15,426 (35.39%)	28,165 (64.61%)	43,591
50–59 years old	20,869 (38.59%)	33,212 (61.41%)	54,081
+60 years old	58,434 (43.20%)	76,823 (56.80%)	135,257
General total	112,168 (40.76%)	162,997 (59.24%)	275,165
**Influenzae virus**	**Female (%)**	**Male (%)**	**Total**
<2 years old	105 (42.86%)	140 (57.14%)	245
2–4 years old	71 (50.71%)	69 (49.29%)	140
5–9 years old	80 (49.69%)	81 (50.31%)	161
10–19 years old	57 (47.90%)	62 (52.10%)	119
20–29 years old	106 (59.55%)	72 (40.45%)	178
30–39 years old	127 (59.07%)	88 (40.93%)	215
40–49 years old	67 (41.88%)	93 (58.12%)	160
50–59 years old	88 (51.46%)	83 (48.54%)	171
+60 years old	170 (45.58%)	203 (54.42%)	373
General total	871 (49.432%)	891 (50.568%)	1762
**Undefined etiological agent**	**Female (%)**	**Male (%)**	**Total**
<2 years old	4752 (42.28%)	6486 (57.71%)	11,238
2–4 years old	3074 (44.89%)	3774 (55.11%)	6848
5–9 years old	2392 (44.76%)	2952 (55.24%)	5344
10–19 years old	2619 (50.56%)	2561 (49.44%)	5180
20–29 years old	5404 (51.11%)	5169 (48.89%)	10,573
30–39 years old	7537 (44.38%)	9444 (55.62%)	16,981
40–49 years old	8863 (40.60%)	12,962 (59.40%)	21,825
50–59 years old	10,664 (41.42%)	15,081 (58.58%)	25,745
+60 years old	36,630 (46.10%)	42,816 (53.90%)	79,446
General total	81,935 (44.73%)	101,245 (55.27%)	183,180

## Data Availability

The data can be obtained at info.gripe.fiocruz.br (accessed on 1 November 2021).

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
