# Peer review of "COVID-19 Underreporting in Brazil among Patients with Severe Acute Respiratory Syndrome during the Pandemic: An Ecological Study"

_diagnostics, 2022, doi:10.3390/diagnostics12061505_

Round 1

Reviewer 1 Report

This study aimed to describe the prevalence and the profile of SARS patients in Brazil, with the purpose of describing the possible underreporting of the SARS-CoV-2 infection. To do so, this study used the data from the fiocruz electronic platform using SINAN data and SIVEP-Gripe surveillance information system.

This is an interesting ecological study comparing the number of SARS patients with COVID-19, influenzae and undefined cause of the 2020/2021 period to the ones of 2015-2019 period to assess an estimation of the underreport rate of SARS-CoV-2 infection in Brazil in SARS patient.

The graphics are presented in a way that information may be biased, and should be improved to represent more clearly the results, see my comments below.

The “Table1” of the study, answering the first aim of description of the prevalence is necessary.

The discussion section is for now focused in detailing how bad the sanitary crisis has been managed by Brazilian Governement, and it is necessary to explain it. However, the discussion section should discuss the result of the paper! Compare the methodology used to calculate the “underreported” cases/deaths to what is usually used in the litterature, compare the findings of the paper with what has been found elsewhere, discuss the results according to the age, and try to find explanation, check bias of the study, and so on.

For the figures:

Please replace the number of weeks by the month/year or at least include some time points to make the read easier (now the reader need to calculate the number of week and guess when is october 2016, or march 2017).

Also, be careful by the scale, as sometimes the font size is not the same (ex: figure 2, the size of the number of the week is different for figure 2.A and 2.B)

figure 3 : present one  figure with 2015-2021 period of time, but after : present only the 270-330 period in order to have the covid period. That will enable to see better the difference between sars, sars by covid, sars by influ and sars without etiologic agent.

In figure 3 : please use SARS total for the blue line, to ensure that the reader understand that this is the total SARS case/death. Also, at some weeks, it seems that the total SARS does not correspond to the SARS covid19+ sars influ + sars without etiologic agent (example: between weeks 270 and 280 in figure 3B). If you present a figure covering only the COVID-19 period, it might help

Figure 4 and 5: please correct the title stating “possible” underdiagnosis. Why do you have only data for 2020? It is difficult to see this result after the graphs including 2021.

Q1: Except for influenza, is there any other reason to have a SARS?

Q2 : in what extent the fiocruz electronic platform using SINAN data have all the SARS syndrome registered? I mean, is there also a possible underreporting of SARS in the electronic system?

Q3: in the methods section: please detail how you had the data from influenza (it is in the abstract but not the main text)

Q4: The figure 4 and 5 represent the number or possible underdiagnosed COVID-19 deaths/cases per age and sex. I f understand quite easily why the number of cases possibly underdiagnosed is growing with age and really high after 60 years, I have more problem with the figure 5. The number of underreported COVID-19 SARS would be higer among those with 2-9 years. Do you have an explanation? Are theses results the same, whatever the geographical region? (If you do the check, you could add the results in supplementary material for example)

Q4: I am not sure what the current table1 adds to the paper. I would add a general table 1 at the beginning of the results section, with the% of male/female  according to Brazilian region, not according to the the age group. And it would be interesting to have the % of case/death according to the SUS/non-SUS health system. This would also help to answer your aim l86: describe the prevalence and the profile of sars-cov-2 patient

Q5: harmonize within the manuscript : undefined cause or undefined etiologic agent

L34 : add “comparing to 2015-2019” instead of “the  other  years  in  the  period”,mor more clarity

Discussion: maybe a paragraph indicating that this study focuses on underreporting among those with SARS , but the underreporting rate is probably much higher when considering all population. (Especially in Brazil where testing is not free, and quite restricted).

Also, as you state l71 to 75, during the first week of the covid19 pandemic, each suspected case were tested, it was not the case after that period. This might impact your results and should be discussed.

Also, please compare the methodology used to calculate the “underreported” cases/deaths to what is usually used in the litterature, compare the findings of the paper with what has been found elsewhere, discuss the results according to the age, and try to find explanation, check bias of the study, and so on.

In the abstract, you state a lot of impressive %, but I thuink that the results stratified on the age group should be taken with caution.

Reviewer 2 Report

Thank you for giving me the opportunity to read and comment a report “COVID-19 Underreporting in Brazil among Patients with severe Acute Respiratory syndrome: A First-Year Report”, by Lima T.M., et al.

In the reviewed manuscript, the possible underreporting of the SARS-CoV-2 in Brazil has been evaluated.

This is a potentially interesting report but at present it is not well written and not suitable for publication.

It is generally accepted that observational studies follow the STROBE and/or RECORD recommendations. Have the authors followed these recommendations? If so, it is advisable to mention it in the "material and methods" section. If these recommendations have not been taken into account, it would be desirable for the authors to modify the study approach and follow the cited recommendations.

Statistical analysis of results is missing. For example, it would be appropriate for categorical variables represented in terms of frequency distribution to be accompanied by their corresponding confidence intervals and for the corresponding statistical test to be applied, for example the chi-square test, if your study population follows a normal distribution, to check if the difference between groups is significant.

The “Results” section is too large. It would be appropriate to summarize the main results. The authors could consider including some table that shows the main data.

It is not necessary to explain the figures in the text again, for that there is already the title of the figure.

The "Results" section should not contain opinions of the authors, just describe the findings aseptically. For example, expressions like “… it is possible to observe a boom phenomenon…(line 127), should be removed.

Finally the format of the bibliography is incorrect. It does not correspond to the recommendations of the journal. Also in the text, the references sometimes appear with a superscript, and sometimes in brackets

Round 2

Reviewer 1 Report

The authors have made several changes, improving the quality of the manuscript. However, the response are not always correct, and  the quality of the redaction is still to be improved. Please consider a review by a medical writer (givin reviews feedback available to this medical writer).

Some basic rules such as : "define the acronym when it first appears" and not again in the text, are not even respected.

Exemple of some problematic points:

The aim of the study is "this study aimed to describe the prevalence and the profile of SARS patients in Brazil, according to the causal factor, age and sex, in a 5-year period, with the purpose of describing the possible underreporting of the SARS-CoV-2 infection."

Please dont use the "causal" worl in an ecological study!

Also, the aim is to describe the prevalence and in the table 1, there is still no description of the prevalence.

The authors stated  that they followed STROBE recommendations, but no changes according to STROBE appear.

At the end of the abstract: "which should be evaluated with caution.", the results should be taken with cautious, not the age!

I suggest to have the manuscript reviewed by a medical writer.

Reviewer 2 Report

The manuscript has improved considerably and is now suitable to publication.
